# Functional Mechanisms of Dietary Crocin Protection in Cardiovascular Models under Oxidative Stress

**DOI:** 10.3390/pharmaceutics16070840

**Published:** 2024-06-21

**Authors:** Sepideh Zununi Vahed, Marisol Zuluaga Tamayo, Violeta Rodriguez-Ruiz, Olivier Thibaudeau, Sobhan Aboulhassanzadeh, Jalal Abdolalizadeh, Anne Meddahi-Pellé, Virginie Gueguen, Abolfazl Barzegari, Graciela Pavon-Djavid

**Affiliations:** 1Kidney Research Center, Tabriz University of Medical Sciences, Tabriz 5165665811, Iran; zununivahed@tbzmed.ac.ir (S.Z.V.); sobhan.aboulhassanzadeh2@mail.dcu.ir (S.A.); 2Université Sorbonne Paris Nord, INSERM U1148, Laboratory for Vascular Translational Science, Nanotechnologies for Vascular Medicine and Imaging, 99 Av. Jean-Baptiste Clément, 93430 Villetaneuse, Franceanne.pelle@univ-paris13.fr (A.M.-P.); virginie.gueguen@univ-paris13.fr (V.G.); abolfazl.barzegari@inserm.fr (A.B.); 3ERRMECe Laboratory, Biomaterials for Health Group, University of Cergy Pontoise, Maison Internationale de la Recherche, I MAT, 1 rue Descartes, 95031 Neuville sur Oise, France; violeta.rodriguez-ruiz@cyu.fr; 4Plateau de Morphologie INSERM UMR 1152 Université Paris Diderot, Université Paris Cité, Bichat Hospital, AP-HP, 46 rue H. Huchard, 75018 Paris, France; olivier.thibaudeau@inserm.fr; 5Immunology Research Center, Tabriz University of Medical Sciences, Tabriz 5165665811, Iran; abdolalizadehj@tbzmed.ac.ir

**Keywords:** crocin, oxidative stress, microRNA, cardiovascular, dietary antioxidants

## Abstract

It was previously reported that crocin, a water-soluble carotenoid isolated from the *Crocus sativus* L. (saffron), has protective effects on cardiac cells and may neutralize and even prevent the formation of excess number of free radicals; however, functional mechanisms of crocin activity have been poorly understood. In the present research, we aimed to study the functional mechanism of crocin in the heart exposed to oxidative stress. Accordingly, oxidative stress was modeled in vitro on human umbilical vein endothelial cells (HUVECs) and in vivo in mice using cellular stressors. The beneficial effects of crocin were investigated at cellular and molecular levels in HUVECs and mice hearts. Results indicated that oral administration of crocin could have protective effects on HUVECs. In addition, it protects cardiac cells and significantly inhibits inflammation via modulating molecular signaling pathways TLR4/PTEN/AKT/mTOR/NF-κB and microRNA (miR-21). Here we show that crocin not only acts as a direct free radical scavenger but also modifies the gene expression profiles of HUVECs and protects mice hearts with anti-inflammatory action under oxidative stress.

## 1. Introduction

Reactive oxygen species (ROS) are byproducts of mitochondrial aerobic respiration and have essential roles as second messengers. ROS transduce intracellular signals through different mechanisms, including ion channels, transcription factors, redox regulation of protein phosphorylation, and cross-linking of the extracellular matrix [1]. Under physiological conditions, the endogenous antioxidant cellular systems detoxify ROS and prevent them from damaging cellular macromolecules and organelles. However, when ROS production exceeds the capacity of the antioxidant defense systems or when antioxidant enzymes are deficient, ROS overproduction disrupts the body’s homeostasis and induces oxidative tissue damage, leading to vascular dysfunction and pro-inflammatory signaling [2]. ROS excess has been shown to participate in the pathogenesis of cardiovascular diseases (CVD) [3], such as atherosclerosis, cardiac thrombosis, myocardial injury, fibrosis, and heart failure [4]. Antioxidant therapeutic strategies have been shown to inhibit oxidative stress-induced cell injury and offer a potential option for the prevention/treatment of CVD [5,6]. However, the limited success of currently available antioxidant agents has led to accelerated growth in the development of novel drugs.

Dietary carotenoids from fruits and vegetables support cells’ function to allow the regulation of ROS formation. A high intake of carotenoids has been reported to be connected with a reduction in CVD risk [7]. However, in some cases, treatment with antioxidants has proven ineffective in clinical trials and sometimes leads to worse outcomes [8]. The control and inhibition of excessive ROS by carotenoids and antioxidant-based formulations are interesting approaches for better clinical efficacy and oxidative stress-derived damages. Among carotenoids, promising compounds are present in *Crocus sativus* (saffron) [9]. The pharmacological properties of saffron as an antioxidant, anticancer, and memory-enhancing factor have been extensively studied in vitro and in vivo [10,11]. In a recent randomized, double-blind, placebo-controlled clinical study (76 patients), dietary supplementation with saffron showed a reduction in cardiovascular risk factors in patients with non-alcoholic fatty liver disease [12]. 

Several studies have shown that saffron extracts and their active ingredients (crocin, picrocrocin, and safranal) can modulate transcription factors, growth factors, and various intracellular signaling pathways [13]. Crocin is a trans-crocetin di-(β-D-gentiobiosyl) ester (PubChem CID: 5281233) and constitutes 6–16% of dried powder of saffron in terms of the origin, culture conditions, and methods of extraction [14,15]. Several studies highlight the potential effect of crocin in the treatment of inflammation and CVD. Crocin reduces oxidative stress, inflammation, apoptosis, and nephrotoxic effects caused by gentamicin in a rat model, and the protective effect was associated with a downregulation of NF-κB/TLR signaling and an upregulation of Nrf-2 and HO-1 [16]. Samarghandian et al. demonstrated that crocin can inhibit the progression of the vascular injury microenvironment in the diabetic rat’s vascular wall [17]. In an LDLR-/- mouse model, intragastric administration of crocin blocked the formation of atherosclerotic plaques by improving hepatic antioxidant capacity [18]. Studies show that crocin as a nutraceutical could normalize lipid metabolism and reduce the accumulation of lipids, thus acting against obesity and disorders linked to dyslipidemia [19]. It was also shown that crocin could decrease lipid peroxidation during ischemia-reperfusion (I/R)-induced oxidative damage in the kidney [8] and skeletal muscle [9] of rats. Crocin was shown to alleviate I/R-induced cardiomyocyte apoptosis both in I/R-induced primary cardiomyocytes and in mouse models [17]. 

We previously showed that aqueous extracts of *C. sativus* protected human endothelial cells under oxidative stress [20]. Here, we studied the effects of crocin pretreatment in vitro and in vivo. The signaling pathways of endogenous antioxidant genes Nrf2/HO1/NQOH1 and TLR4/PTEN/AKT/mTOR/NF-κB signaling pathways that are crucial in cardiac physiology and pathology were investigated in human endothelial cells. Studying carotenoids’ nutrigenomics and their molecular mechanisms could allow us to target their possible future applications.

## 2. Materials and Methods

### 2.1. Reagents

Crocin, 2,2-Diphenyl-1-picrylhydrazyl (DPPH), 2′,7′-dichloro-fluorescein diacetate (DCFH-DA), hydrogen peroxide (H_2_O_2_), 2,2-azobis (2-amidinopropane) dihydrochloride (AAPH), NaCl,3,(4,5-dimethylthiazol-2-yl)2,5diphenyltetrazolium bromide (MTT), ethylenediaminetetraacetic acid (EDTA), NaOH, Tris (hydroxymethyl) aminomethane hydrochloride (Tris-HCl), Triton X-100, sodium deoxycholate, phenylmethylsulfonyl fluoride, Sodium dodecyl sulfate (SDS), protease inhibitor cocktail (P8340), horseradish peroxidase-conjugated secondary antibody were purchased from Sigma-Aldrich (St. Louis, MO, USA). HUVECs (ATCC^®^ CRL-1730, HUV-EC-C™) were purchased from ATCC (LGC, Middlesex, TW11 0LY, UK). Roswell Park Memorial Institute medium (RPMI), fetal calf serum, penicillin-streptomycin-amphotericin and phosphate-buffered saline solution (PBS), 4′,6′-diamidino-2-phénylindole (DAPI), TRIzol^®^ RNA Isolation Reagent, stem-loop RT primer, RT buffer, dNTPs mix, MMLV reverse transcriptase, universal hexamer primer, RNase inhibitor, SYBR Green Master Mix, Pierce™ ECL Western Blotting Substrate (32106), UltraPure™ DEPC-Treated Water were purchased from Thermo-Fisher Scientific SAS (Illkirch, France). Bradford assay from Bio-Rad (Marnes-la-Coquette, France); SDS-PAGE gel and polyvinylidene difluoride membrane were purchased from Millipore (Billerica, MA, USA). Malondialdehyde (MDA), or thiobarbituric acid reactive substances (TBARSs) and Total Antioxidant Status (TAS) were purchased from Randox Laboratories. Serum total cholesterol, HDL, and triglycerides were assessed using enzymatic-colorimetric kits from Pars-azmoon (Tehran, Iran). Primary antibodies anti-Bax (ab77566), anti-NQO1 (ab80588), anti-PTEN (ab137337), anti-HO1 (ab52947), anti-GAPDH (ab37168), anti-Nrf2 (phospho S40) [EP1809Y] (ab76026), anti-MyD88 antibody (ab135693), and anti-AKT1 (ab308381) antibodies were obtained from Abcam (Paris, France). EnVision kit from Dako (Les Ulis, France). 

### 2.2. Crocin and Cell-Free Antioxidant Assay 

#### 2.2.1. DPPH Radical Scavenging Test

Crocin antioxidant activity was determined by the DPPH method as previously described [21]. DPPH is a stable free radical with a deep purple color; after being reduced by an antioxidant, its color turns yellow and its absorbance decreases. The H-transfer reactions from crocin to DPPH were monitored at 517 nm using a UV-Vis spectrophotometer. Briefly, DPPH methanolic solution (100 µM, 3.99 mL) was added to crocin solution (60–3000 µM, 0.1 mL). The mixture was incubated under light protection for 30 min. Crocin anti-radical activity was calculated by comparing the decreasing absorbance of the DPPH in the presence of an antioxidant and the absorbance of the blank DPPH, as follows: (Abs control − Abs sample/Abs control) × 100; Abs control = DPPH solution absorption, and Abs sample = DPPH solution absorption after crocin addition. The concentration of crocin inducing 50% of DPPH activity reduction compared to the control was calculated (IC_50_).

#### 2.2.2. Peroxyl Radical Scavenging Activity (ORAC)

The antioxidant capacity of crocin was assessed by the ORAC method as previously described [22], with slight modifications [23]. Briefly, solutions of fluorescein (4 nM), AAPH (160 mM), and Trolox (0–100 μM) were prepared in PBS. Fluorescein solution (150 μL) and either 25 μL of crocin samples (C), blank (PBS), or standard (T) were added to each well of a 96-well microplate. Finally, pre-heated AAPH at 37 °C (25 μL) was added to the wells to start the reaction. Fluorescence was monitored at 485/528 nm (Ex/Em) every minute for 60 min (3 × 8 measurements/well). ORAC values were stated as Trolox equivalent (in μM) and measured as the slope ratio of relative fluoresce against antioxidants or Trolox at different concentrations (AUC_net_ = AUC_sample_ − AUC_blank_).

### 2.3. Cell Viability and Induction of Oxidative Stress Conditions 

HUVECs were cultured in RPMI medium supplemented with 10% (*v*/*v*) fetal calf serum and 1% penicillin-streptomycin-amphotericin at 37 °C and 5% CO_2_. Cells were detached with trypsin after reaching 80% confluence and seeded at a density of 104 cells/well in 96-well cell culture plates for 24 h. HUVECs were treated with different concentrations of crocin solubilized in PBS (0.125, 0.25, 0.5, 1, 2, and 3 mM) for 24, 48, and 72 h. Then stressors (AAPH or H_2_O_2_) were added to crocin-treated (Cr) and stress control cells (Cr-AAPH, Cr- H_2_O_2_, AAPH, and H_2_O_2_ samples). Cells in basal conditions treated with crocin alone (Cr samples) and non-treated cells in basal conditions (control samples, C_nt_) were included in this study; each experiment was carried out in triplicate. Cell viability was determined by the MTT assay. After incubation, 100 μL of MTT reagent (0.5 mg/mL) was added to each well. Cells were incubated for 4 h in a humidified incubator at 37 °C to allow MTT to be metabolized. After discarding the supernatant, 2-propanol (200 μL/well) was added to dissolve the formazan crystals. The absorbance of the samples was recorded at a 490 nm wavelength using a microplate reader instrument. The percentage of proliferation was calculated with respect to control. The cytotoxicity of the agent was measured as follows: Cytotoxicity (%) = (1− OD of treated cells/OD of control cells) × 100

To induce oxidative stress conditions on HUVECs without cell death, we determined the optimal working concentration of AAPH and H_2_O_2_ (used as stress exogenous inductors). Determination of cell viability under stress conditions (and without crocin) was carried out at 24 and 48 h by MTT assay as described above. The ranges of concentrations for AAPH and H_2_O_2_ were 1.6, 3.12, 6.25, 12.5, 10, and 25 mM, and 50, 100, 200, 400, and 800 μM, respectively. AAPH and H_2_O_2_ IC_50_ were determined as the concentration of each stressor, allowing 50% cell viability. 

### 2.4. Geno-Protective Effect of Crocin 

#### 2.4.1. DAPI Staining and DNA Ladder Assay

HUVECs were cultured for 24 h to be subsequently stressed with either AAPH (5 mM) or H_2_O_2_ (200 µM) in the presence or absence of crocin (500 µM), followed by 48 h of cell incubation. Untreated HUVECs were used as control cells. Then two tests were implemented: 

DAPI staining: HUVECs were fixed with 4% formaldehyde for 15 min and then washed several times with PBS. Cells were permeated with 0.1% (*w*/*v*) Triton X-100 for 5 min, washed again with PBS, and stained with DAPI (200 ng/mL) for 15 min. Staining was verified by a fluorescence microscope. 

DNA ladder assay: Genomic DNA of treated and control cells was extracted and then analyzed by electrophoresis according to the Rahbar et al. protocol [24]. 

#### 2.4.2. Analysis of Cellular DNA Contents by Flow Cytometry: Cell Cycle Arrest Test

The cell cycle phase distribution with cellular DNA contents was carried out using flow cytometry analysis. To this end, the HUVECs were seeded into a six-well plate at a density of 1.0 × 10^6^ cells/mL and treated with the IC_50_ concentration of AAPH (5 mM) and H_2_O_2_ (200 µM) for 48 h in a 5% CO_2_ incubator at 37 °C. After 24 h of incubation, the cells were harvested, washed with PBS, fixed in 70% ethanol, and treated with RNase A (10 mg/mL). The fixed cells were then stained with PI dye, followed by incubation for 10 min at RT. The PI fluorescence of individual nuclei was measured using a flow cytometer, the BD FACS Calibur (Becton Dickinson, Franklin Lakes, NJ, USA).

### 2.5. Effects of Crocin on Mitochondrial Membrane Potential and Cell Signaling

#### 2.5.1. The Mitochondrial Membrane Changes under Oxidative Stress and Crocin Enhancement 

The effect of crocin and stressors on the potential of the mitochondrial membrane (Δψm) was measured using rhodamine-123 (Rh-123). Rh-123 is a lipophilic, cationic fluorescent dye that binds to metabolically active mitochondria. The cells were pre-treated with crocin for 24 h. Then the cells were treated with AAPH (5 mM) and H_2_O_2_ (200 µM) for 12 h. The cells were washed with PBS (pH 7.4) and incubated with 5 µg/mL Rh-123 at 37 °C for 15 min. Afterward, the cells were washed with PBS and imaged by a Cytation™ 5 cell imaging instrument (BioTek, Winooski, VT, USA).

#### 2.5.2. Signaling Regulation by Crocin in HUVEC under Oxidative Stress Conditions

Total RNAs and microRNAs were extracted from crocin-treated (500 µM) and untreated HUVEC cells based on our laboratory protocol. The RNA yield and purity were determined using a NanoDrop ND-1000 spectrophotometer. Reverse transcription and real-time PCR reactions were performed for Nrf_2_, HO1, NQO1, Bax, Bcl-XL, NF-κB, TLR4, MYD88, PTEN, AKT, and mTOR mRNAs as described previously [25,26]. The endogenous controls GAPDH and U6 were used for the normalization of mRNA and miRNA expression levels, respectively. Each reaction was performed in triplicate. Fold change was calculated by the standard 2^−ΔΔCT^ method, where ΔCt = (Ct_target_ gene − Ct_GAPDH_). 

### 2.6. Animal Studies

#### 2.6.1. Animals and Grouping

Twenty male mice were obtained from the experimental animal facilities of Tabriz University of Medical Sciences. All animals were nourished with a standard laboratory diet and water and kept in a controlled environment (20–25 °C, 50% humidity) under a 12 h dark-light cycle. All the experiments were conducted under the guidelines for laboratory animal use and care of the European Community (EEC Directive of 1986; 86/609/EEC). An animal study was carried out following the previously described stress model [27,28] with some modifications. After one week of acclimation, mice were randomly assigned into 4 different groups, receiving diet by gavage. Groups I and II received a normal feed for 14 days, and groups III and IV received crocin (50 mg/kg, gavage). After 14 days, groups II and III were injected (intraperitoneal; IP) with AAPH (5 mg/kg).

#### 2.6.2. Determination of Serum Biochemical Parameters, Total Antioxidant Status, and Lipid Peroxidation

The blood was allowed to clot for 30 min and centrifuged at 2000× *g* for 15 min for a clear separation of serum. Biochemical parameters, including cholesterol (Chol), Malondialdehyde (MDA), Total Antioxidant Status (TAS), high-density lipoprotein (HDL), and triglycerides (Trigs), were studied by colorimetric methods according to the manufacturer’s instructions. Measurement of TAS is based on the incubation of ABTS (2,2′-Azino-di-[3-ethylbenzthiazoline sulphonate]) with a peroxidase (metmyoglobin) and H_2_O_2_ to produce the radical cation ABTS^.^^+^. This product has a relatively stable blue-green color, which is measured at 600 nm. When antioxidants are present in the sample, ABTS^.^^+^ loses its color to a concentration-proportional degree. Lipid peroxidation in serum was determined by thiobarbituric acid reactive substances (TBARS) based on the reaction of MDA with thiobarbituric acid; forming a reaction product that absorbs at 532 nm.

#### 2.6.3. Histopathological Analysis of the Heart Tissue

Mice hearts were excised and washed with saline solution before being fixed in a 3.7% neutral-buffered formalin solution for at least 48 h. Tissue samples were embedded in paraffin and cut into 5-mm thick sections. Hematoxylin-Phloxine-Saffron (HPS) staining was used for morphological evaluation. Labeling of Nrf2 (phospho S40), NQO1, and HO1 antibodies was determined by immunostaining (working dilution: 1/30, 1/300, 1/300, respectively) and revealed by an EndVision kit. Digital images were obtained with a Nano Zoomer 2.0-RC C 10730 (Hamamatsu, Japan).

#### 2.6.4. Western Blot Analysis

Frozen tissues were homogenized on ice in three volumes of 10 mM TrisHCl, 1.5 mM MgCl2, 10 mM KCl containing 1 mM dithiothreitol (DTT), 1 mM of the phosphatase inhibitor, Na3VO4, and a protease inhibitor cocktail. The solution was centrifuged at 5000× *g* for 5 min at 4 °C, and the extracted cytoplasmic proteins were stored at −80 °C. The Bradford assay (Bio-Rad) was used for determining protein concentrations using BSA as the standard. Equal amounts of protein were separated by 10% sodium dodecyl sulfate-polyacrylamide gel electrophoresis (SDS-PAGE) and transferred to polyvinylidene fluoride membranes (0.45 µm pore size). Membranes were blocked with phosphate-buffered saline with 0.05% Tween-20 (PBST) containing 5% nonfat dry milk for 1 h and then incubated at 4 °C overnight with an anti-MyD88 antibody. Membranes were then washed with PBST, incubated with a secondary antibody for 1 h at room temperature, and then detected using an ECL Western Blotting Detection Reagent. Protein levels were normalized to GAPDH. For HUVECs, the process was performed according to the same protocol with modifications. The cell extracts were separated by a 12% SDS-PAGE gel and transferred to a membrane. The membrane was stained with primary antibodies specific to Bax, NQO1, PTEN, HO1, and GAPDH before being incubated with a horseradish peroxidase-conjugated secondary antibody (1:2000). Bands were detected using an ECL Western Blotting Substrate chemiluminescent kit.

### 2.7. Statistical Analysis

Data were expressed as mean ± standard deviation from three independent experiments. Statistical significance between groups was analyzed by a one-way ANOVA followed by Dunnett’s multiple comparisons tests. The significance level was a *p*-value of <0.05.

## 3. Results

### 3.1. Antioxidant Activity of Crocin 

The DPPH method is a single electron transfer-based assay employed for measuring the ability of antioxidants to transfer labile H atoms to radicals [21]. In this assay, antioxidants are evaluated based on their EC50 value (the concentration necessary to reduce 50% of DPPH). DPPH involves a single redox reaction in which the endpoint indicator is the oxidant [29]. Figure 1A illustrates the free radical scavenging activities of crocin assessed by the DPPH assay. The EC_50_ DPPH scavenging activity of crocin was 1000 µM. The ORAC method was used for analyzing crocin antioxidant activity against AAPH. ORAC is a standard assessment based on the kinetic evaluation of a hydrogen atom transfer-type reaction [30]. The reduction of the fluorescence intensity of fluorescein due to the action of AAPH was monitored for 1 h. The antioxidant activity of crocin was assessed as its capacity to inhibit fluorescein fluorescence decay in the presence of AAPH (Figure 1B). The antioxidant activity of crocin was calculated relative to Trolox activity, an antioxidant reference molecule. The results showed that the antioxidant activity of crocin corresponds to 1.13 Trolox equivalents (µM). 

### 3.2. Crocin Protects Endothelial Cells against Exogenous Stress Induced by AAPH and H_2_O_2_ In Vitro

The viability of HUVEC was determined after incubation with different concentrations of crocin (125, 250, 500, 1000, 2000, and 3000 µM) for 24, 48, and 72 h. As shown in Figure 1C, results demonstrated that crocin did not inhibit cell proliferation at concentrations below 1000 µM. Moreover, higher antioxidant concentrations (2000 µM) presented an inhibitory effect on cell growth after 24 h of supplementation; crocin became toxic at 3000 µM. This is not surprising since an apoptotic effect of high crocin concentrations (greater than 3000 µM) has been reported on cancer cells [31]. Thus, high crocin concentrations hamper the normal ROS signaling pathways needed to ensure cell survival. The capacity of crocin to protect HUVEC against stress-induced conditions (during 24 and 48 h) by AAPH and H_2_O_2_ was evaluated. As shown in Figure 1D,E, 5 mM AAPH or 200 µM H_2_O_2_ induced HUVEC death (cell viability ~50%) after 48 h. Crocin (500 μM) protected cells against AAPH (Figure 1F) and H_2_O_2_ (Figure 1G) toxic effects and increased HUVEC viability up to ~80%. Moreover, an apoptotic synergistic action was observed when high concentrations of crocin (≥1000 μM) were mixed with AAPH and H_2_O_2_.

### 3.3. Crocin Protection against HUVEC DNA Damage

Both the DAPI staining and DNA ladder assay are methods that measure DNA damage as a marker of exposure to genotoxic agents and the evaluation of the genoprotective capacities of compounds. These techniques were used to investigate the effects of H_2_O_2_ and AAPH on DNA damage and evaluate crocin capacity to counteract stress-induced toxicity (Figure 2A–F). As shown in Figure 2A–F, crocin protected cells against apoptosis induced by AAPH and H_2_O_2_ (Figure 2C,D). ROS induced by AAPH and H_2_O_2_ showed a marked increase in DNA fragmentation compared with controls. Conversely, crocin protected cells against AAPH and H_2_O_2_ ROS toxicity and decreased DNA damage (Figure 3A). 

### 3.4. Influence of Crocin on the Signaling Pathway of HUVEC under Oxidative Stress 

#### 3.4.1. Crocin Carries out Its Antioxidant Activity through the Nrf2/HO1/NQO1 Pathway

Nuclear factor-erythroid 2-related factor 2 (Nrf2), a key transcription factor, controls antioxidant defenses, including the antioxidant enzymes heme oxygenase-1 (HO-1) and NAD(P)H quinone oxidoreductase 1 (NQO1) [32]. HO-1, a stress response protein, is induced in response to a variety of oxidative challenges and pathological stimuli to serve as a cytoprotective factor. It mediates the anti-inflammatory effects [33] and has a central role in cardiovascular protection [34]. Increased levels of Nrf2, HO1, and NQO1 genes were observed in cells under AAPH-induced oxidative stress as a defense mechanism against cell injury (Figure 3B,C). However, treatment with crocin significantly diminished the mRNA and protein levels, probably due to crocin’s scavenging activity to neutralize intracellular ROS, preventing cells from employing any cellular antioxidant mechanisms to combat free radicals. In response to the H_2_O_2_-induced ROS, significant upregulation of Nrf2 expression was observed; however, levels of HO-1 and NQO1 did not significantly change (Figure 3B,C). One possible reason is that under H_2_O_2_-induced stress, Nrf2 induces catalase activity rather than HO-1 and NQO1 expression. However, the treatment of cells with crocin alone did not lead to the modulation of the expression levels of the genes mentioned above.

#### 3.4.2. Crocin Exerts Its Cytoprotective and Survival Potential via the AKT/mTOR Signaling Pathway on HUVEC Cells

The association of AKT and the protein kinase mammalian target of rapamycin (mTOR) pathways is required for cell proliferation and survival. mTOR, a serine/threonine protein kinase, regulates cell motility, growth, proliferation, protein synthesis, autophagy, and survival. The mTOR pathway has a regulatory function in cardiovascular physiology and pathology in the unstressed and stressed-out heart. Moreover, mTORC1 is essential for the protection of cardiac structure and function [35]. Cellular AKT and mTOR levels are regulated by PTEN. PTEN, a critical upstream regulator of Akt phosphorylation, directly interacts with Akt, preferentially binding to its phosphorylated form (p-Akt) and dephosphorylating it, thereby inactivating Akt. Loss or downregulation of PTEN leads to increased levels of p-Akt due to the removal of PTEN’s inhibitory effect [36,37]. Current evidence suggests that measuring PTEN levels can provide insights into the potential status of p-Akt. In the present study, AAPH and H_2_O_2_ stress induction almost blocked the expression of AKT and mTOR, while crocin (500 µM) significantly increased the expression of these genes at mRNA and protein levels in the presence of AAPH and H_2_O_2_ in comparison to the stressed-out cells (Figure 3D,E). Down-regulation of PTEN was observed in crocin-treated cells. However, up-regulation of PTEN in H_2_O_2_ stimulation could de-activate AKT and lead cells to apoptosis (Figure 3F,G). We hypothesized that the observed increase in cell survival is due to the elevated levels of AKT and mTOR as well as diminished levels of PTEN in response to crocin action.

#### 3.4.3. Crocin Inhibits HUVEC Apoptotic Genes Induced by AAPH and H_2_O_2_

To evaluate mitochondrial-related apoptosis, expression levels of Bax and Bcl-XL were detected in the crocin-treated cells. Significantly, elevated levels of pro-apoptotic Bax and diminished levels of anti-apoptotic Bcl-XL were seen upon AAPH and H_2_O_2_ stimulation. Moreover, reduced levels of Bax and increased levels of Bcl-XL genes were observed after 24 h crocin (500 µM) pretreatment when compared to the respective stress-out AAPH and H_2_O_2_ cells (Figure 3F,G). 

### 3.5. AAPH and H_2_O_2_ Induce G0/G1 Phase Arrest and Apoptosis Phase

Cell cycle analysis with cellular DNA content was performed by flow cytometry. As shown in Figure 4, when cells were exposed to 5 mM AAPH and 200 µM H_2_O_2_ for 48 h, treated cells had a higher % of cells under G1 (33.92 and 14.44% for AAPH and H_2_O_2_, respectively) compared to control (0.02%). However, co-treatment of stressed cells with crocin resulted in a significant decrease in Sub G1 (Figure 4). 

### 3.6. Mitochondrial Protection by Crocin 

Rhodamine 123 (Rh123) staining was utilized for the real-time monitoring of mitochondrial membrane potential (ΔΨm) changes in response to oxidative stimuli. Rh123 is a widely used and sensitive fluorescent dye for staining and detecting ΔΨm in living cells. It is a cationic dye that accumulates in the mitochondrial matrix in a membrane potential-dependent manner in cells, enabling the study of mitochondrial function and health. When mitochondria are energized, Rh123 fluorescence is quenched due to its electrophoretic accumulation in the matrix. Conversely, mitochondrial depolarization leads to a release of Rh123 from mitochondria and an increase in fluorescence [38,39].

Under oxidative stress, there is a significant loss of ΔΨm concurrent with cell death. Additionally, oxidative stress can lead to alterations in mitochondrial membrane permeability and structure, affecting the biophysical properties of mitochondrial membranes and impairing the function of respiratory enzymes, ultimately contributing to a decrease in ΔΨm [40]. AAPH can induce oxidative stress, leading to a decrease in mitochondrial membrane potential, likely due to mitochondrial uncoupling, lipid peroxidation, and increased ROS production [41]. Untreated HUVECs showed high fluorescence, indicating a polarized mitochondrial membrane (Figure 5). The AAPH-treated cells and H_2_O_2_ showed disruption of H pumps, and fluorescence intensity was also decreased. The notable results are that crocin modulates ATP synthesis via direct antioxidant activity and possibly other signaling pathways. 

### 3.7. Effects of Crocin on Stress-Induced Heart Injury in Mice

#### 3.7.1. Protective Effects of Crocin on the Histology of Mice Heart

The schematic representation of animal study is indicated in Figure 6A. Histological staining revealed a cardiac morphology arranged into myocardial bundles, showing slight differences among the animal groups (Figure 6B). However, HE staining did not allow the identification of stress lesions on the heart tissues. To evaluate the real free radical damage inflicted by oxidative stress-AAPH action and the protective effect of crocin, immunological staining of Nrf2 was carried out. Nrf2, as explained above, is a pivotal transcription factor that regulates the gene expression of intracellular antioxidants and detoxifying enzymes that neutralize ROS [42]. Under pathological conditions, Nrf2 is liberated from the Keap1-Nrf2 complex and translocated to the nucleus, stimulating antioxidant response gene expression. Nrf2 immunohistological staining of different groups (Control, AAPH, and AAPH/Crocin) showed two different positive stainings: a dark brown nuclear color (yellow arrows) and a positivity found in the muscle fibers. This cytoplasmic stain was observed in all groups and was particularly concentrated within the intercalated discs. It could be a reflection of the high-stress state of heart tissues, as expected due to normal contractile heart movements. The nuclear translocation of Nrf2 was observed slightly more strongly in the AAPH group and zones 1 and 2 of the diagram, with higher nuclear staining when compared to the control and crocin-treated groups (Figure 6B). These results could correspond to a protective effect of crocin treatment on cardiomyocytes under AAPH-induced oxidative stress. 

#### 3.7.2. Serum Biochemical Parameters

The redox state of the sera was evaluated by TAS and MDA measurements (Figure 6C). AAPH stress conditions induced an increase in TAS and MDA but also in triglyceride CHOL and HDL. Under crocin treatment, decreased levels of serum TAS, triglyceride, and HDL cholesterol were observed when compared to AAPH-treated mice groups. However, no significant differences were observed between the groups (Figure 6C). 

#### 3.7.3. Crocin Exerts Its Antioxidant Activity via TLR4/MYD88 Receptors in Mice

Toll-like receptors (TLRs) have a serious role in the activation of the innate immune response. The activation of the TLR4 signaling pathway is induced by damage-associated molecular pattern molecules (DAMPs) under oxidative stress conditions [43]. TLR4 and its adapter protein, myeloid differentiation primary response protein (MYD88), induce the nuclear factor-κB (NF-κB) that leads to pro-inflammatory cytokine production and cardiac impairment. miR-21 has an essential role in oxidative-stress-dependent endothelial dysfunction and is deregulated in the heart and vasculature under CVD conditions [44]. miR-21 plays an important role in tissue injury, inflammation, and heart pathogenesis, making it a potential therapeutic target [45]. Moreover, miR-21 serves to enhance NF-κB activation [46]. Given the critical roles of TLR4, MYD88, NF-κB (RelA subunit of the transcription complex), and miR-21 in inflammation, we evaluated the effect of crocin on oxidative AAPH-induced inflammation in the mice’s heart tissue. The results indicated that crocin could significantly decrease the elevated levels of TLR4, MYD88, NF-κB, and miR-21 in the stressed mice hearts when compared to the AAPH-induced stress group (Figure 7A–C).

## 4. Discussion

Oxidative stress has a crucial role in the pathophysiology of CVDs such as hypertension, atherosclerosis, and heart failure, as well as disorders such as inflammation and diabetes [47,48,49]. Evidence indicates that the vascular endothelium is susceptible to oxidative stress-induced injury due to the extreme production of ROS [50]. Thus, efforts aimed at diminishing ROS generation and increasing antioxidant bioavailability in patients with CVDs could have a significant potential impact. Antioxidant therapy targeting oxidative stress-related CVDs has been studied in humans [51,52]; however, its molecular mechanisms are not entirely clear. Consequently, understanding the molecular mechanisms of potent natural compounds with antioxidant properties is a prerequisite for the development of anti-oxidative stress therapy [53]. 

Crocin antioxidant activity was evaluated by DPPH and ORAC tests, confirming the antioxidant capacities of crocin. However, the EC50 of crocin activity (1000 µM) was 10 times lower compared to the antioxidant activity of the aqueous extract of saffron (100 µM) [25], suggesting that other hydrophilic components of saffron participate in the antioxidant activity of natural saffron. In the present study, crocin’s antioxidant and cytoprotective effects against AAPH and H_2_O_2_-induced HUVEC cell injury were confirmed by DPPH, ORAC, DAPI, and DNA ladder assays. Our observations are consistent with previous studies [25,54,55]. Hosseinzadeh et al. showed that an aqueous extract of Crocus sativus stigmas and crocin protected mice’s organs against the genotoxic effect of methyl methanesulfonate-induced DNA damage [56]. Moreover, it was reported that saffron aqueous extract exerts a protective effect against genotoxin-induced oxidative stress in mice [57].

It has been shown that the AKT/mTOR signaling pathways could be blocked by oxidative stress to reverse cell longevity and metabolism and cause cell death; therefore, loss of AKT and mTOR leads to apoptosis [58,59]. Consistent with these previous studies, our results showed a decrease in the AKT/mTOR pathway during AAPH and H_2_O_2_ application. However, crocin could increase mTOR transcriptional levels and AKT at both translational and transcriptional levels when compared to cells under stress, thereby increasing cell survival. In agreement with these results, PTEN was decreased in crocin-treated cells. PTEN is a well-known inhibitor of AKT phosphorylation. PTEN down-regulation strengthened AKT signaling and cell survival in AAPH-induced oxidative stress in HUVEC cells. In H_2_O_2_ and AAPH-induced-stressed cells, an increased apoptosis rate was observed in comparison to controls through increased PTEN and BAX and decreased Bcl-XL expression levels. These findings suggested a possible role for crocin in encouraging cell survival via activation of PTEN/AKT/mTOR signaling and inhibition of mitochondria-mediated apoptosis in HUVECs. Since AAPH induces ROS in the cell membrane and cytoplasm but not directly in mitochondria, it seems that the molecular signaling pathway induced by AAPH is different from that induced by H_2_O_2_.

TLR signaling in cardiomyocytes contributes to the pathogenesis of serious cardiac conditions, particularly inflammation. It is reported that up-regulated TLR4 in cardiomyocytes triggers an inflammatory response, which reduces cardiomyocyte contractility [60]. Moreover, TLR4 amplifies heart failure after long-term myocardial infarction [61]. Given its critical role in TLR signaling, MyD88 also mediates the myocardial innate immune response and injury [62]. Both TLR4 and MyD88 may contribute to myocardial inflammation and infarction after I/R via NF-κB activity [34]. Increased mRNA and protein levels of MYD88 in the stressed-out hearts of mice indicated that AAPH caused inflammation through the TLR4/MYD88/NF-κB pathway [63]. On the one hand, NF-κB activity is necessary for miR-21 induction, on the other hand, miR-21 serves to enhance NF-κB activation. In the present study, crocin was shown to significantly decrease the mRNA levels of TLR4, MYD88, NF-κB, and miR-21, indicating that crocin exerted an anti-inflammatory effect against oxidative stress in mice. 

## 5. Conclusions

In conclusion, in the present study, we investigated the role of crocin as an intracellular scavenger and an antioxidant factor. Crocin protects endothelial cells against oxidative stress via activation of Nrf2. Moreover, it exerted cell survival, anti-inflammatory, and anti-apoptosis activity by modulating PTEN/AKT/mTOR, TLR4/MYD88/NF-κB/miR-21, and mitochondrial Bax/Bcl-XL signaling, respectively, in both cell culture and animal models of oxidative stress (Figure 8).

## Figures and Tables

**Figure 1 pharmaceutics-16-00840-f001:**
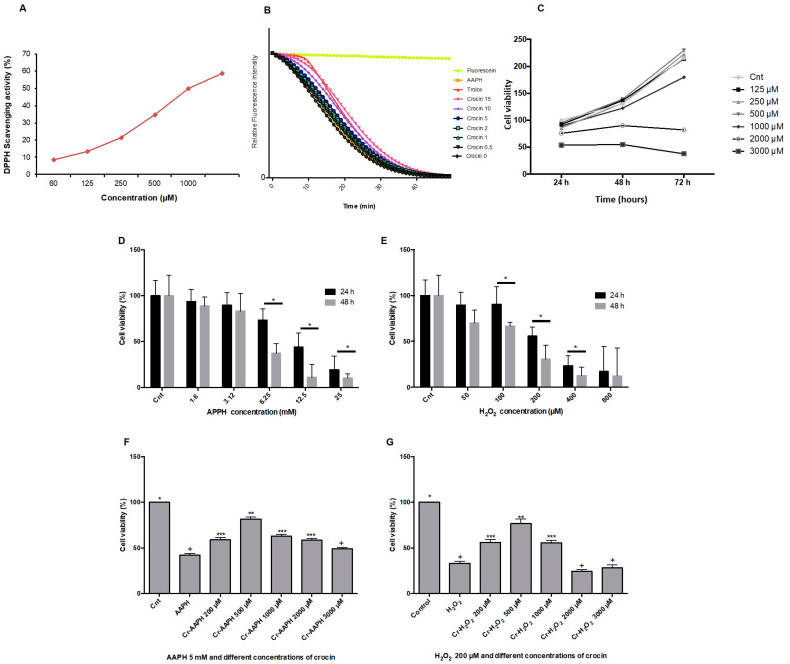
Antioxidant activity of crocin. (**A**) The antioxidant capacity of crocin was evaluated by DPPH. The IC_50_ (concentration required to inhibit 50% of DPPH radicals) of crocin was ~1000 µM. (**B**) Kinetics of fluorescence decay measured by the oxygen radical antioxidant capacity (ORAC) assay. (**C**) HUVEC cell viability determination by MTT assay at 24, 48, and 72 h after treatment with 125, 250, 500, 1000, 2000, and 3000 µM of crocin. (**D**,**E**) Effects of different concentrations of AAPH (**D**) and H_2_O_2_ (**E**) on cell viability. (**F**,**G**) Effects of different concentrations of crocin and AAPH 5 µM (**F**,**G**) H_2_O_2_ 200 µM on cell viability. Crocin 500 μM protects cells against AAPH and H_2_O_2_ toxic effects and increases HUVEC viability. The data show the mean values ± SD of at least three independent experiments. (**D**,**E**) * represents statistical significance *p* < 0.05, ** *p* < 0.01, *** *p* < 0.001 compared to control (Cnt). (**F**,**G**) Levels not connected by the same symbol (+, *, **, ***) are significantly different (*p* < 0.05).

**Figure 2 pharmaceutics-16-00840-f002:**
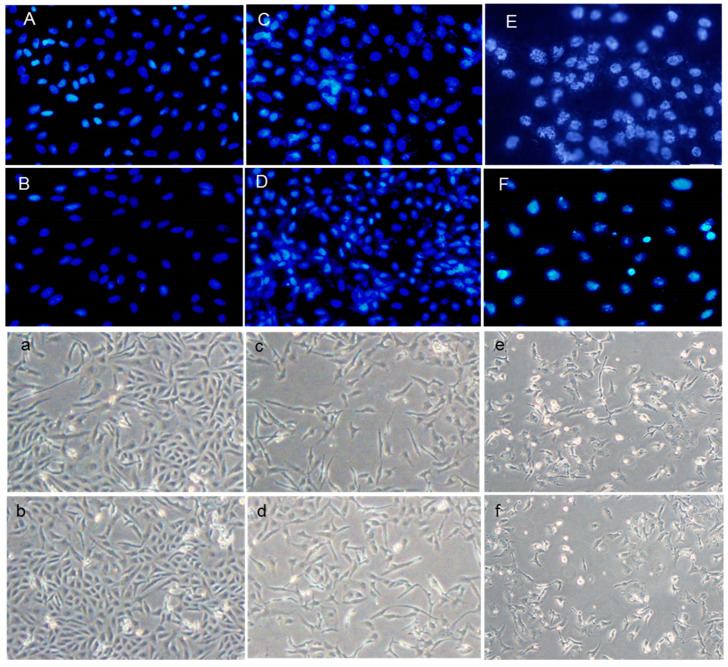
Protective effect of crocin on HUVEC DNA damage. (**A**–**F**) DAPI staining (magnification 40×) and (**a**–**f**) light microscope (magnification 20×). (**A**,**a**) control cells; (**B**,**b**) crocin-treated cells; (**C**,**c**) AAPH+crocin-treated cells; (**D**,**d**) H_2_O_2_+crocin-treated cells; (**E**,**e**) AAPH-treated cells; and (**F**,**f**) H_2_O_2_-treated cells.

**Figure 3 pharmaceutics-16-00840-f003:**
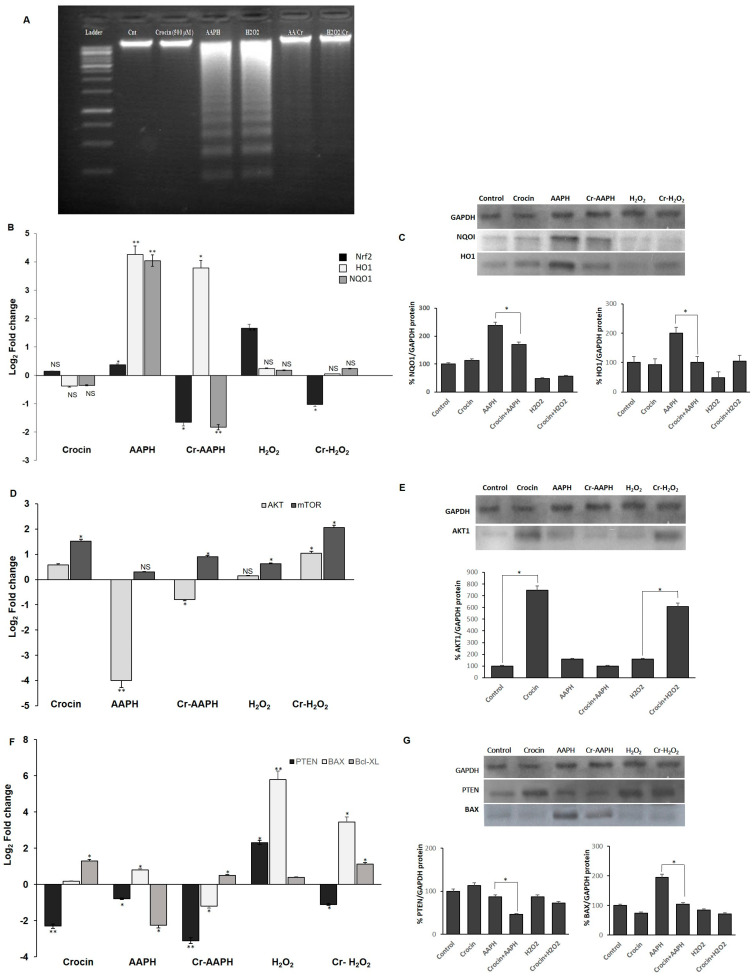
Antioxidant effects of crocin on HUVEC cells. (**A**) Crocin protects HUVECs DNA against AAPH and H_2_O_2_, as evaluated by the DNA ladder assay. (**B**,**C**) Crocin exerts antioxidant properties through modulating Nrf2, HO1, and NQO1 (**B**) mRNA and (**C**) protein levels. Effects of crocin on HUVEC cell survival and apoptosis. (**D**,**F**) mRNA and (**E**,**G**) protein levels of AKT, mTOR, PTEN, BAX, and BCL-XL in HUVEC-treated cells. The 2^−ΔΔCt^ method was used to calculate the relative expression (fold change) between sample groups. Relative expression is indicated as the mean (SD) of the log (_2_) fold change. GAPDH was used as an endogenous control. Cr: crocin. NS: not significant. The data show the mean values ± SD of at least three independent experiments. * *p* < 0.01, ** *p* < 0.001.

**Figure 4 pharmaceutics-16-00840-f004:**
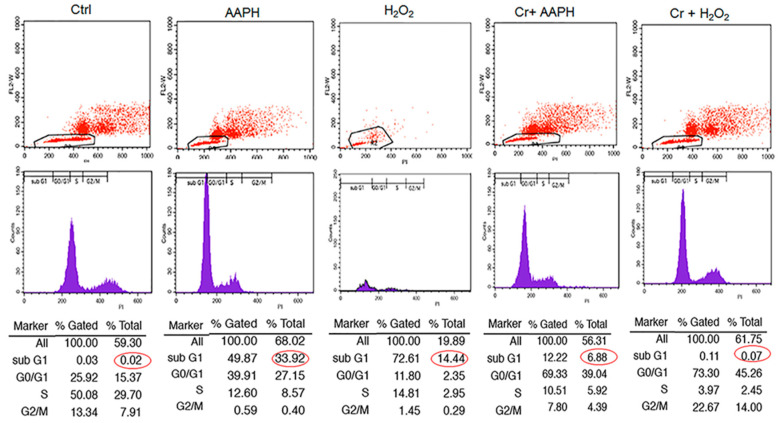
The effect of crocin on different phases of the cell cycle under oxidative stress. HUVECs were treated with IC_50_ concentrations of AAPH and H_2_O_2_ for 48 h, stained with PI, and analyzed by flow cytometry. The figure shows the quantitative distribution of HUVECs in different phases of the cell cycle in the untreated cells, the cells treated with AAPH and H_2_O_2_, and the cells co-treated with crocin and stressors. Red circles indicate apoptotic cells in sub G1 population. Crocin treatment could decrease the number of apoptotic cells in sub G1 population compared to AAPH and H_2_O_2_ groups in flow cytometric analysis.

**Figure 5 pharmaceutics-16-00840-f005:**
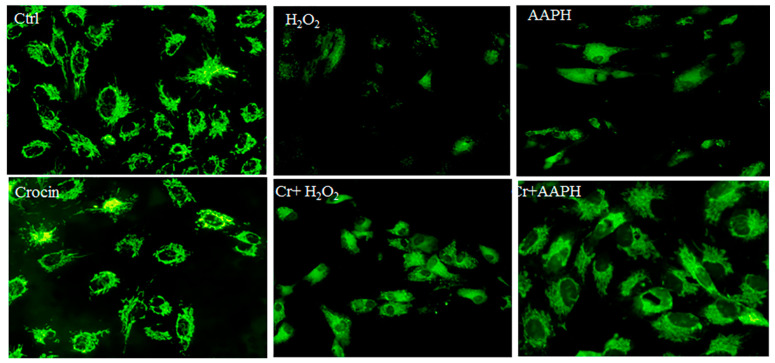
Rhodamine 123 staining for mitochondrial membrane potential. The figure represents the untreated HUVECs, which show high fluorescence, indicating a polarized mitochondrial membrane. AAPH- and H_2_O_2_-treated cells show low fluorescence (magnification 20×). It means that oxidative stress could affect ATP synthesis, and crocin co-treated modified the mitochondrial membrane potential. Cr: Crocin.

**Figure 6 pharmaceutics-16-00840-f006:**
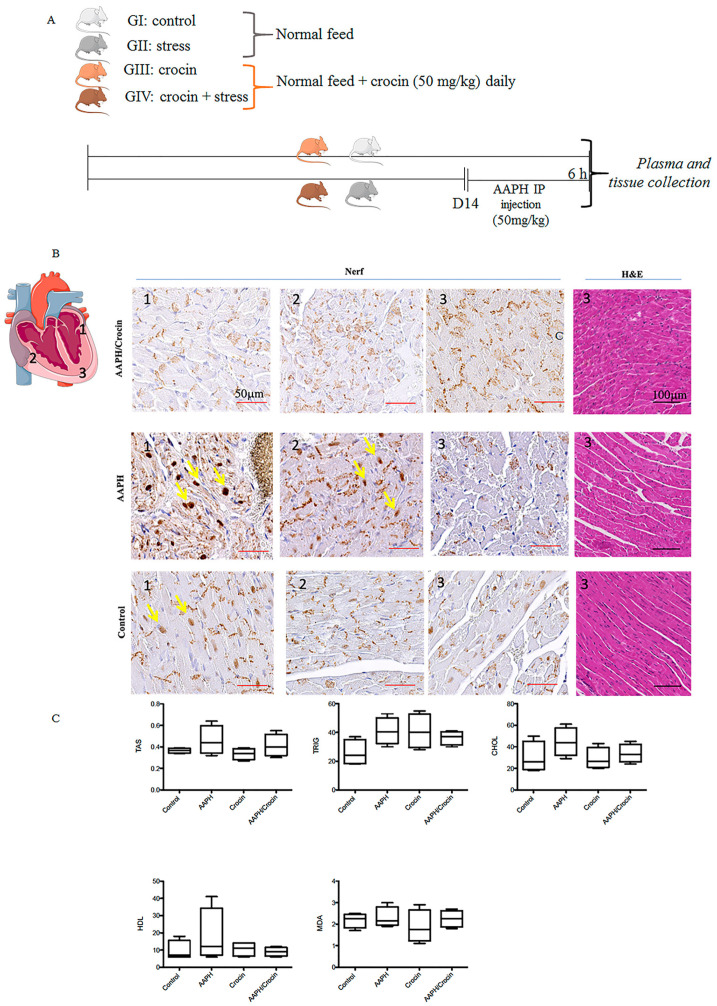
Crocin’s protective effects on stressed-out mice. (**A**) Experimental setting: four groups of mice receiving diet by gavage. GI and GII received a normal feed, and in GIII and GIV, crocin (50 mg/kg). After 14 days, the GII and GIII were injected intraperitoneally with AAPH (5 mg/kg), and the samples were collected after 6 h of stress treatment. (**B**) Immunohistological Nrf2 staining in three different zones of the mouse heart under AAPH-induced oxidative stress: non-treated (GII) or treated with crocin (GIV) and control (GI); the yellow arrow indicates nuclear positive staining; the positive basal labeling is attributed to muscle fiber positive staining concentrated within the intercalated disc. (**C**) Serum biochemical parameters TAS: Total Antioxidant Status, Trig: triglyceride; HDL: high-density lipoprotein, Chol: cholesterol, MDA: Malondialdehyde. The data show mean values ± SD of at least three independent measurements of serum parameters.

**Figure 7 pharmaceutics-16-00840-f007:**
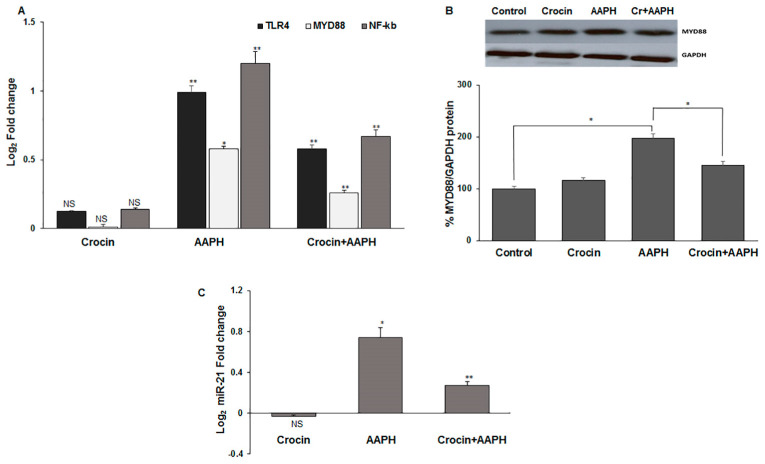
The signaling pathway is affected by crocin in mice hearts (**A**) mRNA levels of TLR4, MYD88, and NF-κB, (**B**) MYD88 protein, and (**C**) miR-21 transcriptional levels in stress-induced mice hearts. The 2^-ΔΔCt^ method was used to calculate the relative expression (fold change) between sample groups. GAPDH and U6 were used as internal controls, respectively, for mRNA and protein, as well as miR-21 normalization. Fold change is indicated as the mean (SD) of log (2). The data show mean values ± SD of at least three independent experiments. * *p* < 0.01, ** *p* < 0.001. NS: not significant.

**Figure 8 pharmaceutics-16-00840-f008:**
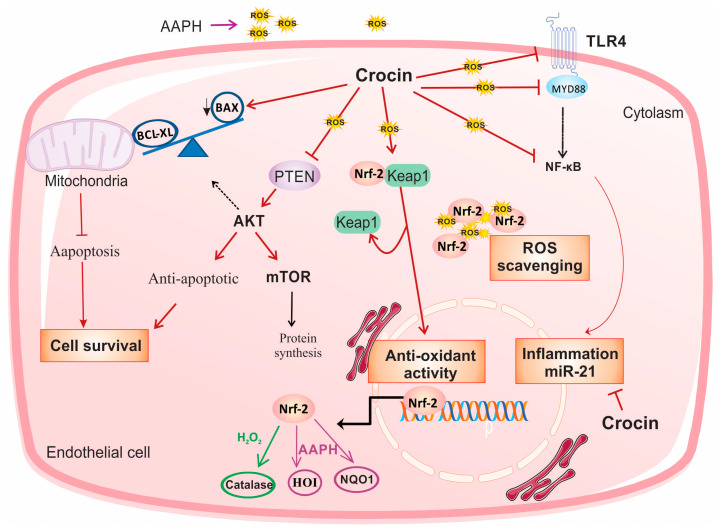
Suggested schematic illustration of the mechanism by which crocin could induce cell survival and exert anti-oxidative and anti-inflammatory effects in stressed cardiomyocytes. In response to the treatment of AAPH and H_2_O_2_, crocin represents antioxidant and anti-inflammatory properties through activation of Nrf2 and inhibition of NF-κB, respectively. Activation of Nrf2 is required for the cellular antioxidant activity of crocin. In response to the AAPH-induced stress, crocin acts as a free radical scavenger, and it somehow modulates Nrf2/HO1/NQO1 expression. Inhibition of NF-κB, either through TLR4 and MYD88 or other related signaling, decreases transcription of pro-inflammatory genes and miR-21 and overall declines inflammation. Moreover, crocin hinders PTEN from surviving cells against oxidative stress. A decreased level of PTEN significantly induces AKT expression at both transcriptional and translational levels, which stimulates mTOR signaling and leads to cell proliferation and survival. Moreover, crocin induces anti-apoptotic factors to block mitochondrial-induced cell apoptosis. Since AAPH induces ROS in the cell membrane and cytoplasm but not directly in mitochondria, the molecular signaling pathway induced by AAPH may be different from H_2_O_2_; therefore, it is rational to observe differences in gene expression levels in response to crocin treatment in AAPH and H_2_O_2_-induced stressed-out cardiomyocytes (see text). mTOR: protein kinase mammalian target of rapamycin, TLR4: tool-like receptor 4, Nrf-2: nuclear factor-erythroid 2-related factor 2, HO-1: heme oxygenase-1, NQO1: NAD quinone oxidoreductase 1, ROS: reactive oxygen species.

## Data Availability

The datasets used and/or analyzed during the current study are available from the corresponding author upon reasonable request.

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
