# Peer review of "Functional Mechanisms of Dietary Crocin Protection in Cardiovascular Models under Oxidative Stress"

_pharmaceutics, 2024, doi:10.3390/pharmaceutics16070840_

Round 1

Reviewer 1 Report

Comments and Suggestions for Authors

The article of S. Zununi Vahed and coauthors “Functional mechanisms of dietary crocin protection in cardiovascular models under oxidative stress” describes the hypothetical mechanism of the effect of Crocin during oxidative stress. The present study is important for understanding the action of the dietary antioxidant.

There is no doubt that this article should be published in the journal, but the manuscript needs to be finalized.

1.      line 246 …polyvinylidene fluoride membranes - indicate the pore diameter

2.      Figure 1 (D, E, F, and G)…The figure shows significant results but does not indicate how many repetitions were used and what p is equal to.

3.      Figure 3… The figure shows significant results but does not indicate how many repetitions.

4.      In signal transduction, it is not Akt that is important, but phosphor-Akt, which is activated or inhibited by various stimuli.

It is necessary to repeat this experiment and indicate the ratio of p-Akt to Akt.

5.      Figure 6 … the significance and number of experiments are not indicated

6.      Figure 7… The figure shows significant results but does not indicate how many repetitions.

If the authors consider the content of PTEN, it is necessary to study mitophagy in their further studies.

Reviewer 2 Report

Comments and Suggestions for Authors

This paper talks about saffron crocin's ability to destroy free radicals, to modify gene expression profiles of human umbilical vein endothelial cells under in vitro oxidative stress, and to protect mice heart against inflammation under cellular stressors in vivo. 

The authors claim that research on crocin is scarce, which is surprising. The medical virtue of saffron is well known and many medications are based on saffron, and even crocin extracts. Maybe, they want to emphasize the specificity of the study they conducted, in which case, it is unfair (and not scientifically sound) to just ignore and bypass the other research. Please review the introduction and include these missing references. This is a must for me to accept the paper.

The authors made particular effort in presenting their methods. They were careful in choosing their providers, at least those that were disclosed in the paper. Minor edits are needed as some sections are put at the same level as subsections. For example, section #2.3 is not similar to section #2.2, which is in bold. It is misleading as the reader expect the 2.3 to be at the same level as 2.2.2. There are a few other sections in a similar situation.

Combine 2.4 & 2.5 into just one section (both address DNA) and 2.6 & 2.7 (both address physico-chemical aspect) into another single section to have even size of section (not a must).

For section 2.10, it would be helpful if you list out the concerned data because not all data were statistical.

Sections 3.2 & 3.3 should be in bold like the others. 

Section 3.5: include the highlighted or encircled numbers in the text to support your statement. Also, add the percent of decrease.

Section 3.6: the title is misleading. No where in the text you gave the MEASRED potential. And it is missing in the label of Fig 5 too. Please state that measured potential or change the title. Also, the text has nothing to do with membrane potential. It is not straight forward. First, include a reference about the fluorescence, then give more in depth explanation.

Line #505 talks about [53], which is not in the list of reference at all.

I have no doubt that this will be a great paper with a broad impact after the authors address the comments above and the comments from the other referees. Thank you.

Round 2

Reviewer 1 Report

Comments and Suggestions for Authors

Authors

“Thank you for your insightful comments regarding the importance of phospho-Akt in signal transduction. PTEN, the upstream regulator of Akt phosphorylation, is a dual lipid and protein phosphatase that interacts directly with Akt and preferentially binds to its phosphorylated form (p-Akt, activated form). This interaction allows PTEN to dephosphorylate and inactivate Akt. Moreover, loss/downregulation of PTEN leads to increased levels of p-Akt (1, 2). Therefore, based on the current evidence, by measuring PTEN levels, we can infer the potential status of p-Akt. Moreover, p-Akt directly activates mTORC1, which then facilitates the full activation of Akt through mTORC2 (3, 4).”

 Reviewer. 

I agree with the authors on this issue, but this point needs to be clearly stated in the article.

In addition, it is necessary to describe the anti-Akt antibodies that you used, that they include both phospho-Akt and non-phospho. In addition, it is necessary to describe the anti-Akt antibodies that you used, that they include both phospho-Akt and non-phospho, and explain why you are using PTEN and not using phospho-Akt, insert the appropriate links that the authors showed in the answers to the questions.

Comments on the Quality of English Language

 Minor editing of English language required
